# Aqueous humour proteins and treatment outcomes of anti-VEGF therapy in neovascular age-related macular degeneration

Yusuke Arai[1], Hidenori Takahashi[1,2,3]*, Satoru Inoda[1], Xue Tan[2,3], Shinichi Sakamoto[1], Yuji Inoue[1,2], Yujiro Fujino[3], Hidetoshi Kawashima[1], Yasuo Yanagi[4,5,6]

1 Department of Ophthalmology, Jichi Medical University, Shimotsuke-shi, Tochigi, Japan, 2 Department of Ophthalmology, Graduate School of Medicine, the University of Tokyo, Bunkyo-ku, Tokyo, Japan, 3 Japan Community Health Care Organization Tokyo Shinjuku Medical Center, Shinjuku-ku, Tokyo, Japan, 4 Department of Ophthalmology, Asahikawa Medical University, Asahikawa-shi, Hokkaido, Japan, 5 Medical Retina, Singapore National Eye Centre, Singapore, Singapore, 6 Medical Retina, Singapore Eye Research Institute, Singapore, Singapore

* takahah-tky@umin.ac.jp

**Data Availability Statement:** All data set files are available from the Figshare database. (https://doi.org/10.6084/m9.figshare.7126922.v1)

## Abstract

We aimed to construct a better model for predicting treatment outcomes of anti-vascular endothelial growth factor therapy for neovascular age-related macular degeneration (nAMD) using the concentrations of aqueous humour proteins at baseline and during treatment. From the data of 48 treatment-naïve nAMD eyes that received intravitreal ranibizumab *pro re nata* for up to 12 months, we used the aqueous humour concentrations of C-X-C motif chemokine ligand 1 (CXCL1), CXCL12, CXCL13, interferon-γ-induced protein 10, monocyte chemoattractant protein 1 (MCP-1), C-C motif chemokine ligand 11, interleukin 6 (IL-6), IL-10, and matrix metalloproteinase 9 (MMP-9). After stepwise regression, multivariate analysis was performed to identify which predictors were significantly associated with best-corrected visual acuity (BCVA) changes and the number of injections. The results demonstrated that besides male sex (β coefficient = −0.088, $P$ = 0.040) and central retinal thickness (β coefficient = 0.00051 per μm, $P$ = 0.027), MCP-1 (β coefficient = 0.44, $P$ < 0.001) and IL-10 (β coefficient = −0.16, $P$ = 0.033) were significantly correlated with baseline BCVA. Additionally, high MCP-1 at baseline (β coefficient = −0.20, $P$ = 0.015) and low CXCL13 at baseline (β coefficient = 0.10, $P$ = 0.0054) were independently associated with better BCVA change at 12 months. High MMP-9 at the first injection (β coefficient = 0.56, $P$ = 0.01), CXCL12 at the third injection (β coefficient = 0.10, $P$ = 0.0002), and IL-10 at the third injection (β coefficient = 1.3, $P$ = 0.001) were predictor variables associated with the increased number of injections. In conclusion, aqueous humour protein concentrations may have predictive abilities of BCVA change over 12 months and the number of injections in *pro re nata* treatment of exudative nAMD.

## Introduction

Many biomarker studies have been conducted to investigate the associations between the intraocular concentrations of anti/pro-angiogenic factors and treatment outcomes in

**Funding:** This work was supported by a KAKENHI grant from the Japan Society for the Promotion of Science, Grant Number 15K10899.

**Competing interests:** Dr Takahashi received lecturer's fees from Kowa Pharmaceutical, Novartis Pharmaceuticals, Bayer Yakuhin, and Santen Pharmaceuticals, and grants from Bayer Yakuhin and Novartis Pharma, outside this work. Dr Inoue received lecturer's fees from Kowa Pharmaceuticals, Novartis Pharmaceuticals, Bayer Yakuhin, and Santen Pharmaceuticals outside this work. Dr Kawashima received lecturer's fees from Kowa Pharmaceutical, Novartis Pharmaceuticals, and Santen Pharmaceuticals outside this work. Dr Yanagi received lecturer's fees and grants from Santen Pharmaceuticals outside this work. He is an advisory board member for Bayer Pharmaceuticals and a consultant for Santen Pharmaceuticals. Dr Arai, Dr Tan, Dr Inoda, Dr Sakamoto, and Dr Fujino declare no potential conflict of interest. This does not alter our adherence to PLOS ONE policies on sharing data and materials.

neovascular age-related macular degeneration (nAMD), with a hope to improve prognostic accuracy. For example, several studies have demonstrated that aqueous humour protein concentrations may help differentiate good responders from poor responders in the setting of anti-vascular endothelial growth factor (VEGF) treatment [1–3]; however, the prognostic value of such aqueous humour protein concentrations in clinical care remains under debate. Besides the information obtained at presentation, additional information acquired after/during initial dosing and maintenance therapy may help predict long-term treatment outcomes; therefore, the importance of assessing disease activity after the loading dose has been highlighted in recent studies [4–11]. For example, visual and anatomical outcomes up to 12 weeks are associated with long-term treatment outcomes such as visual acuity and treatment frequency [4–11]. Some studies [12,13], including ours [13], measured aqueous humour proteins during treatment and demonstrated their change after treatment. As such, the investigation of aqueous humour proteins at baseline and after the loading phase may be useful to better predict treatment outcome. Furthermore, information is also lacking on whether cytokine concentrations in the aqueous humour are correlated with visual acuity (VA). Thus far, no studies have used aqueous humour samples at baseline and later visit(s) to investigate the relationship between these concentrations and VA at baseline and treatment outcome.

We previously investigated changes in the concentrations of multiple aqueous humour proteins during the induction phase, i.e. before and after the initial 3 monthly consecutive injection phase, of ranibizumab treatment in patients with nAMD, using initial samples [13]. Among C-X-C motif chemokine ligand 1 (CXCL1), CXCL12, CXCL13, interferon-γ-induced protein 10 (IP-10), monocyte chemoattractant protein 1 (MCP-1), C-C motif chemokine ligand 11 (CCL11), interleukin 6 (IL-6), IL-10, and matrix metalloproteinase 9 (MMP-9), selected based on previous clinical and animal studies [12–14], we found that pro-inflammatory cytokines are elevated in the aqueous humour of nAMD patients, while MMP-9 levels are decreased [13]. We also found that after the induction phase, inflammatory cytokine levels in the aqueous humour were strongly suppressed, while MMP-9 levels are increased. No cytokine was significantly different between type 1 and 2 choroidal neovascularization (CNV).

In the present study, we built on our previous work and aimed to clarify whether we could construct a better model for predicting treatment outcomes of anti-VEGF therapy for nAMD by using the concentrations of aqueous humour proteins at baseline and during treatment. We used stepwise regression followed by multiple regression analysis to investigate the relationship between the concentrations of various cytokines and VA at baseline and at 2 and 12 months after treatment, as well as the number of injections during the first year of anti-VEGF monotherapy with a standardised *pro re nata* treatment regimen.

## Materials and methods

### Study design and approval

We used data from our previous prospective study [13]. The present study was conducted in accordance with the Declaration of Helsinki. Written informed consent was obtained from all patients. Institutional Review Board approval was obtained from the Japan Community Health Care Organization, Tokyo Shinjuku Medical Center.

### Data collection

The study procedure has been described previously [13]. In brief, we analysed 48 eyes of 48 treatment-naïve patients with nAMD (typical nAMD, 21 eyes; polypoidal choroidal vasculopathy [PCV], 27 eyes) with a mean age of 72.9 years. The patients received three consecutive intravitreal injections of ranibizumab monthly from November 2010 to August 2012 at the Japan Community Health

Care Organization Tokyo Shinjuku Medical Center. Patients with other ocular diseases, including glaucoma and anterior uveitis, were excluded. The comparison of cytokine concentrations before the first and third intravitreal ranibizumab injections was published elsewhere [13].

Samples of aqueous humour (approximately 0.2 mL) were collected by manual aspiration into disposable syringes through a clear corneal paracentesis under an operating microscope immediately before each injection. The samples were transferred immediately to sterile tubes, snap-frozen, and stored at −80˚C until required.

The concentrations of the following aqueous humour proteins were determined using a multiplex cytokine assay (Procarta® Cytokine Assay Kit; Affymetrix, Inc., Santa Clara, CA) according to the manufacturer's instructions: CXCL1, IP-10, CXCL12, CXCL13, MCP-1, CCL11, IL-6, IL-10, and MMP-9. We imputed the concentration of half the minimal detection limit when the concentration was lower than the detection limit. Because aqueous protein concentrations have been reported to not reliably reflect vitreous cytokine concentrations [14], 5 vitreous humour samples were examined preliminarily for the presence of the 9 proteins. The presence of all studied proteins in the vitreous humour was confirmed, and the mean (95% confidence interval) concentrations were 5.7 (2.3–13.5), 87 (47–159), 370 (240–570), 12 (5–28), 567 (436–737), 4.4 (3.0–6.4), 4.6 (1.7–12.5), 0.20 (0.10–0.36), and 1.6 (0.2–11.2) pg/mL for CXCL1, IP-10, CXCL12, CXCL13, MCP-1, CCL11, IL-6, IL-10, and MMP-9, respectively. We confirmed that dye angiography did not affect the measurements (data not shown). The concentration of VEGF, as measured using an enzyme-linked immunosorbent assay (Quanti-kine® Human VEGF Immunoassay; R&D Systems, Inc., Minneapolis, MN), was available in 29 samples, as has been reported elsewhere [13]. We first subjected the samples to multiplex analysis and then the leftover was subjected to ELISA. Therefore, we were able to measure VEGF concentrations in 29 (60%) cases prior to anti-VEGF therapy.

## Clinical examinations

To diagnose typical nAMD, PCV [15], and retinal angiomatous proliferation, both fluorescein and indocyanine green angiography were performed routinely for nAMD cases at our institution, with some exceptions (e.g. contraindications due to drug allergy, liver dysfunction, or recent cerebrovascular event) [13]. For the measurement of axial length, A-mode ultrasonography (UD-6000; Tomey Corp., Aichi, Japan) was used. B-mode ultrasonography (UD-6000) was used To examine posterior vitreous detachment (PVD), as previously described [16–21], and the eyes were either categorised into the PVD group or the without PVD group.

All patients received 3 monthly ranibizumab (0.5 mg/0.05 mL) injections followed by *pro re nata* injections [9]. Each patient underwent a best-corrected VA (BCVA) test with a 5 M Landolt C VA chart (i.e. decimal VA chart) and spectral domain optical coherence tomography (OCT; Cirrus HD-OCT Model 4000; Carl Zeiss Meditec AG, Oberkochen, Germany) at the initial and each follow-up visit. The patients were followed up on a monthly basis, and retreatment was given based on the qualitative assessment of OCT; if any persistent or recurrent intra-retinal or subretinal fluid and/or clinically detectable retinal haemorrhage was observed, an injection was recommended. We did not include BCVA loss in the retreatment criteria.

BCVA was converted to a logarithm of minimum angle of resolution (logMAR) values for statistical analysis. Greatest linear dimension (GLD) was determined based on fluorescein angiography. Disease duration was based on the patient's symptoms. Central retinal thickness (CRT) was determined from the distance between the inner limiting membrane and Bruch's membrane. Central choroidal thickness (CCT) was defined as the distance from Bruch's membrane to the choroid-scleral border. The values for thickness were determined manually at the foveal centre using the OCT calliper function.

## Statistical analysis

Statistical analysis was performed using JMP Pro software version 11.2.0 (SAS Institute, Cary, NC). A two-sided paired $t$-test was performed to compare the changes of BCVA and CRT at baseline and at 12 months. Associations between baseline factors, cytokine concentrations, and visual and anatomical outcomes were examined with Pearson's correlation coefficient after a normal distribution was confirmed with Shapiro-Wilk's $W$-test or Spearman's rho for categorical variables. Stepwise variable selection (using minimum Bayesian information criterion, increasing the number of variables) was performed using baseline BCVA, BCVA change at 12 months, number of ranibizumab injections, and CRT and CCT (at baseline and change at 2 months) as response variables, and disease type, disease duration (months), sex, age, axial length (mm), PVD, GLD (μm), CRT (μm), CCT (μm), BCVA, and each log concentration of aqueous humour proteins as predictor variables. The concentrations of aqueous humour proteins were log-transformed because of the lognormal distribution of these variables. To select the predictor variables in multiple regression analysis, we used the non-fixed stepwise method due to its objectivity and to avoid the risk of multicollinearity [22]. We then used multiple regression analysis after stepwise variable selection. $P < 0.05$ was considered to be significant.

## Results

### BCVA and central retinal thickness over 1 year of treatment

After receiving 3 monthly intravitreal injections of ranibizumab, all 48 eyes received *pro re nata* intravitreal injections of ranibizumab. The changes in BCVA and CRT over 12 months are shown in Fig 1. BCVA and CRT improved by 0.12 logMAR units and 180 μm after treatment, respectively (both $P < 0.01$).

### Factors associated with baseline VA

At baseline, worse BCVA was associated with increased CRT (r = 0.43, $P = 0.020$), IP-10 (r = 0.44, $P = 0.016$), MCP-1 (r = 0.54, $P = 0.0025$), and IL-6 (r = 0.44, $P = 0.018$) in univariate analysis (Table 1). Multiple regression analysis after stepwise variable selection demonstrated that besides male sex (β coefficient = −0.088, i.e. regression coefficient was −0.088 logMAR for men [as 1] vs women [as 0], $P = 0.040$) and CRT (β coefficient = 0.00051 logMAR per μm, $P = 0.027$), logMCP-1 (β coefficient = 0.44, i.e. baseline BCVA worsened by 0.44 logMAR per logMCP-1, $P < 0.001$) and logIL-10 (β coefficient = −0.16, $P = 0.033$) were independently associated with baseline BCVA (Table 1).

### Factors associated with BCVA at 12 months

To investigate the factors associated with treatment outcome at 12 months, we used demographic characteristics, OCT parameters at baseline and at 2 months, and aqueous humour biomarker concentrations at baseline and at 2 months. An improvement of BCVA was associated with worse BCVA at baseline (r = -0.31, $P = 0.031$) and high MCP-1 at baseline (r = −0.29, $P = 0.042$) in univariate analysis (Table 2). Stepwise selection demonstrated that the parameters at baseline and 2 months were independently associated with the change in BCVA. Among the baseline demographic factors, the presence of PCV (β coefficient = -0.095, $P = 0.00076$) and smaller GLD (β coefficient = 4.3e−5, $P = 0.00068$) were associated with better BCVA change at 12 months (Table 2). Among the cytokine concentrations at baseline and at 2 months, high MCP-1 at baseline (β coefficient = −0.20, $P = 0.015$) and low CXCL13 at baseline (β coefficient = 0.10, $P = 0.0054$) were also independently associated with better BCVA change

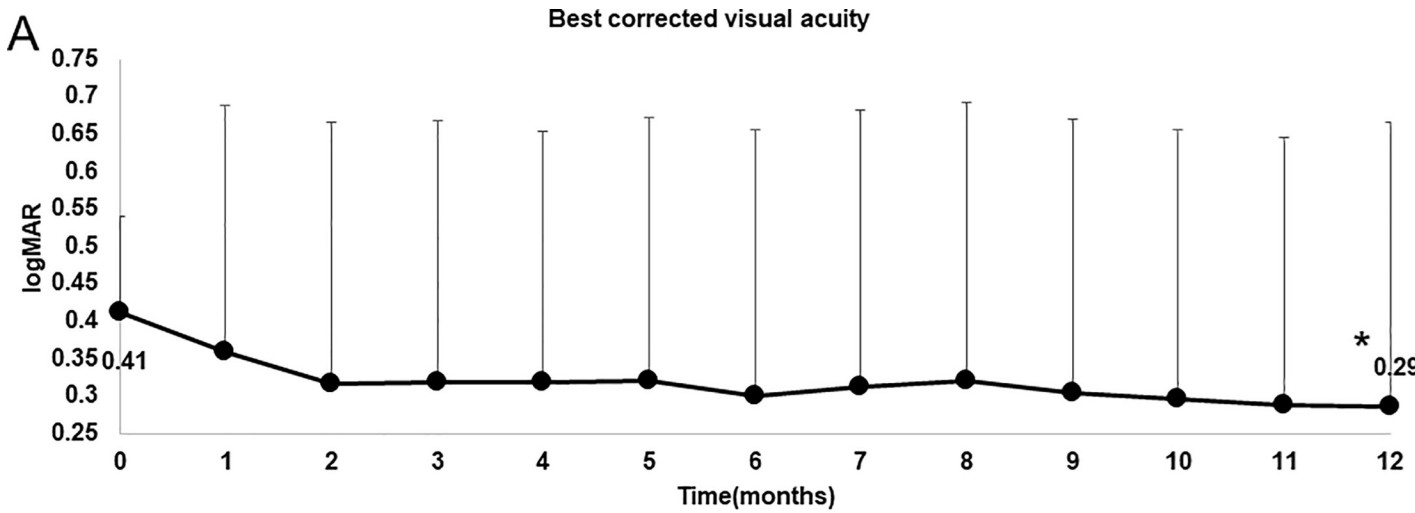

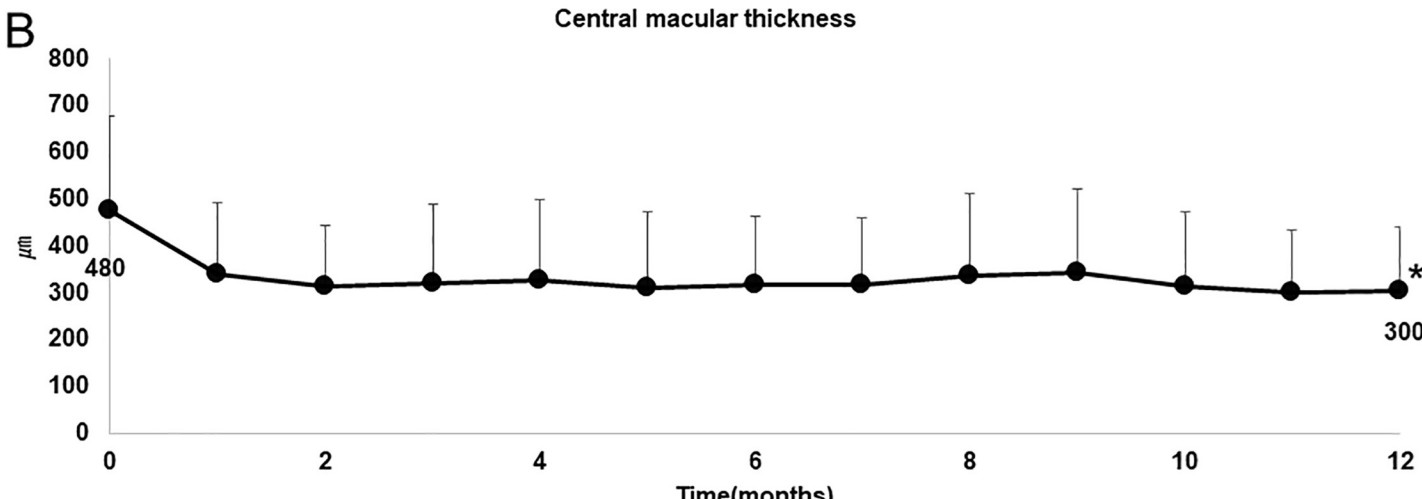

**Fig 1. Best-corrected visual acuity (BCVA) and central retinal thickness (CRT) over 1 year of treatment in neovascular age-related macular degeneration (nAMD).** Forty-eight eyes with nAMD received 3 monthly intravitreal injections of ranibizumab followed by *pro re nata* intravitreal injections of ranibizumab. **a**: Last mean BCVA (0.29) was better than initial mean BCVA (0.41) (paired *t* test. $P = 0.0028$). **b**: Last mean CRT (300 μm) was better than initial mean CRT (480 μm) (paired *t* test. $P < 0.0001$).

at 12 months. Overall, 69% of the BCVA change over 12 months could be explained by the combination of these factors ($R^2 = 0.69$) (Table 2).

## Factors associated with the number of injections within 12 months

Factors associated with the number of ranibizumab injections were also investigated in univariate analysis, which demonstrated that only CXCL12 at the third injection (r = -0.40, $P = 0.0045$) was associated with the number of injections among the baseline characteristics, baseline OCT parameters, baseline cytokines, and 3-month cytokine concentrations. However, stepwise selection and multiple regression analysis demonstrated that the predictor variables initial BCVA (β coefficient = −1.5, $P = 0.04$), high MMP-9 at the first injection (β coefficient = 0.56, $P = 0.01$), CXCL12 at the third injection (β coefficient = 0.10, $P = 0.0002$), and IL-10 at the third injection (β coefficient = 1.3, $P = 0.001$) were associated with the increased

**Table 1. Factors associated with baseline visual acuity.**

| | BCVA (logMAR) | |
| --- | --- | --- |
| | Univariate | Multivariate |
| Age (years) | −0.070 (0.71) | |
| Sex (male) | −0.20 (0.058) | **−0.088 (−0.170 to −0.007, 0.040)***  |
| BCVA (logMAR) | | |
| Disease type (PCV) | 0.028 (0.79) | |
| Duration of disease (months) | −0.043 (0.82) | |
| GLD (μm) | 0.25 (0.19) | |
| PVD (+) | 0.024 (0.82) | |
| Axial length (mm) | −0.063 (0.74) | |
| CRT (μm) | **0.43 (0.020)*** | **0.00051 (0.00007 to 0.00094, 0.027)*** |
| CCT (μm) | 0.15 (0.42) | 0.00083 (−0.00016 to 0.00181, 0.11) |
| VEGF (pg/mL) | 0.13(0.50) | |
| CXCL1 (pg/mL) | 0.14 (0.46) | |
| IP-10 (pg/mL) | **0.44 (0.016)*** | |
| CXCL12 (pg/mL) | 0.20 (0.30) | |
| CXCL13 (pg/mL) | 0.27 (0.16) | |
| MCP-1 (pg/mL) | **0.54 (0.0025)*** | **0.44 (0.20 to 0.68, <0.001)*** |
| CCL11 (pg/mL) | 0.18 (0.35) | |
| IL-6 (pg/mL) | **0.44 (0.018)*** | |
| IL-10 (pg/mL) | −0.13 (0.50) | **−0.16 (−0.31 to −0.02, 0.033)*** |
| MMP-9 (pg/mL) | 0.044 (0.82) | |
| $R^2$ | | 0.49 |

Univariate: Pearson's correlation (for continuous variables) and Spearman's rho (for categorical variables). R (*P* value).

Multivariate analysis was performed after stepwise variable selection (BIC, forward method). β Coefficient (95% confidence interval, *P* value).

*$P < 0.05$.

BCVA: best-corrected visual acuity; CCL11: C-C motif chemokine ligand 11; CCT: central choroidal thickness; CRT: central retinal thickness; CXCL1: C-X-C motif chemokine ligand 1; CXCL12: C-X-C motif chemokine ligand 12; CXCL13: C-X-C motif chemokine ligand 13; GLD: greatest linear dimension; IL-6: interleukin 6; IL-10: interleukin 10; IP-10: interferon-γ-induced protein 10; MCP-1: monocyte chemoattractant protein 1; MMP-9: matrix metalloproteinase 9; PCV: polypoidal choroidal vasculopathy; PVD: posterior vitreous detachment; VEGF: anti-vascular endothelial growth factor.

number of injections ($R^2 = 0.38$). Importantly, clinical and demographical parameters excluding aqueous humour cytokines were not associated with the number of injections (Fig 2).

## Discussion

Anti-VEGF therapy has achieved unprecedented success in the treatment of nAMD [23]. Angiogenesis and vascular hyperpermeability in nAMD are caused mainly by VEGF; however, the long-term management of exudative nAMD with anti-VEGF therapy remains challenging [24,25], partly because of significant contributions from other factors [26]. As we and others have shown, eyes with nAMD have elevated concentrations of several cytokines/chemokines (e.g. MCP-1 [27] and C-X-C motif chemokines such as IP-10 [28]) in the aqueous humour. Such molecules partly modulate the activity of CNV directly through pro- and anti-angiogenic activity or by recruiting monocytes, which positively and negatively regulate inflammation by

**Table 2. Factors associated with best-corrected visual acuity change at 12 months.**

| | BCVA (logMAR) | |
|---|---|---|
| | **Univariate** | **Multivariate** |
| Age (years) | −0.13 (0.49) | |
| Sex (male) | 0.14 (0.10) | |
| BCVA (logMAR) | −0.31 (0.031) | |
| BCVA change at 2 months (logMAR) | 0.67 (<0.0001) | **0.86 (0.63 to −1.08, <0.0001)*** |
| Disease type (PCV) | −0.088 (0.30) | **-0.095 (0.044 to 0.146, 0.0008)*** |
| Duration of disease (months) | −0.017 (0.93) | |
| GLD (μm) | 0.28 (0.14) | **4.3e-5 (2.0e-5 to 6.5e-5, 0.0007)*** |
| PVD (+) | −0.11 (0.20) | |
| Axial length (mm) | −0.14 (0.48) | |
| CRT (μm) | 0.040 (0.84) | |
| CRT change at 2 months (μm) | 0.34 (<0.019) | |
| CCT (μm) | 0.18 (0.36) | |
| CCT change at 2 months (μm) | −0.13 (0.36) | |
| VEGF (pg/mL) | −0.18 (0.36) | |
| CXCL1 (pg/mL) | −0.25 (0.18) | |
| CXCL12 (pg/mL) | −0.15 (0.43) | |
| CXCL13 (pg/mL) | −0.018 (0.93) | **0.10 (0.03 to 0.17, 0.0054)*** |
| IP-10 (pg/mL) | −0.048 (0.80) | |
| MCP-1 (pg/mL) | −0.29 (0.042) | **−0.20 (−0.34 to −0.05, 0.015)*** |
| CCL11 (pg/mL) | −0.14 (0.45) | |
| IL-6 (pg/mL) | −0.11 (0.58) | |
| IL-10 (pg/mL) | 0.12 (0.53) | |
| MMP-9 (pg/mL) | −0.024 (0.90) | |
| $R^2$ | | 0.69 |

Univariate analysis: Pearson's correlation (for continuous variables) and Spearman's rho (for categorical variables). R (*P* value).

Multivariate analysis was performed after stepwise variable selection (BIC, forward method). β Coefficient (95% confidence interval, *P* value).

*P < 0.05.

BCVA: best-corrected visual acuity; CCL11: C-C motif chemokine ligand 11; CCT: central choroidal thickness; CRT: central retinal thickness; CXCL1: C-X-C motif chemokine ligand 1; CXCL12: C-X-C motif chemokine ligand 12; CXCL13: C-X-C motif chemokine ligand 13; GLD: greatest linear dimension; IL-6: interleukin 6; IL-10: interleukin 10; IP-10: interferon-γ-induced protein 10; MCP-1: monocyte chemoattractant protein 1; MMP-9: matrix metalloproteinase 9; PCV: polypoidal choroidal vasculopathy; PVD: posterior vitreous detachment; VEGF: anti-vascular endothelial growth factor.

inducing production of other angiogenic/inflammatory cytokines, such as VEGF, IL-6 [29], and IL-10 [30]. Importantly, in addition to cytokines and chemokines, tissue proteases such as MMP-9 [31], and intracellular adhesion molecules such as intercellular adhesion molecule 1 and vascular cell adhesion molecule 1 [32–34], are also overexpressed in CNV and accelerate CNV growth [35]. Such molecules are also detected in the aqueous humour.

In this study, we investigated the associations between the concentrations of 9 aqueous humour proteins (CXCL12, CXCL13, IL-10, IP-10, IL-6, MCP-1, CCL11, CXCL1, and MMP-9) from 48 treatment-naïve eyes with nAMD at baseline and at 2 months after the first ranibizumab injection and BCVA, CRT, CCT, and the number of injections.

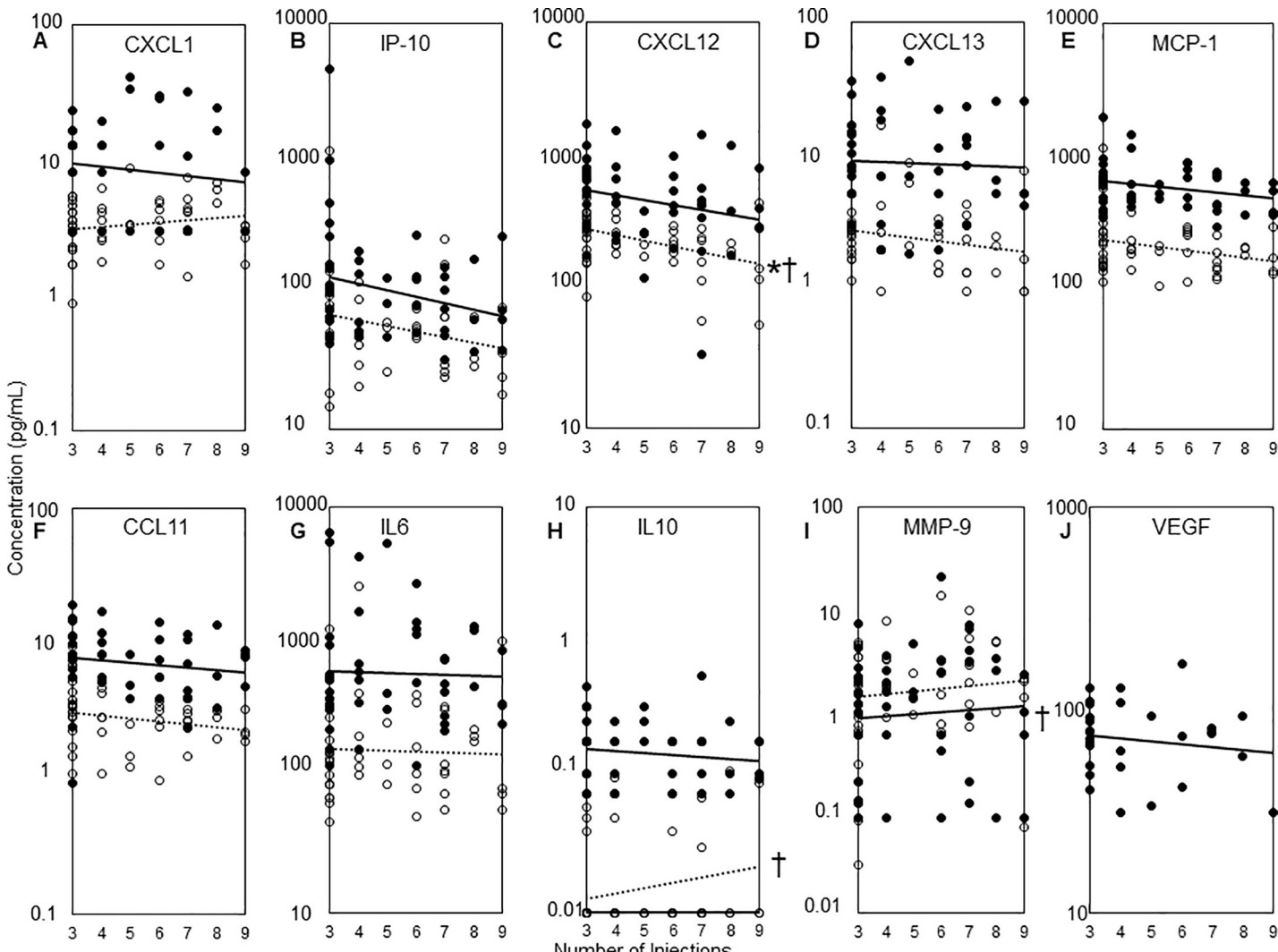

**Fig 2. Correlation between the number of injections and cytokine concentrations. a**: CXCL1, **b**: IP-10, **d**: CXCL13, **e**: MCP-1, **f**: CCL11, **g**: IL-6, and **j**: VEGF concentrations did not significantly correlate with the number of injections. **c**: CXCL12 concentration at the third injection was correlated with the number of injections in univariate analysis ($P = 0.0045$) and predicted the number of injections in multivariate analysis ($P = 0.0002$). **h**: IL-10 concentration at the third injection was a predictor variable that was positively associated the number of injections in multivariate analysis ($P = 0.001$). **i**: MMP-9 concentration at the first injection predicted the number of injections in multivariate analysis ($P = 0.01$).

Overall, regarding ocular imaging biomarkers, our results confirmed the findings of previous studies. Here, eyes with a thick choroid were relatively resistant to anti-VEGF therapy, similar to previous reports [36]. As expected, the change inBCVA at 12 months was better in eyes with a small GLD and bad initial BCVA.

## VA and cytokines

A previous study revealed that MCP-1 was not associated with initial BCVA, but IL-10 was positively correlated with the change of VA [37]. The results of the present study demonstrated that initial BCVA was better in eyes with a lower concentration of MCP-1 and higher IL-10 at baseline.

MCP-1 is a major chemoattractant for inflammatory monocytes and aggravates inflammation [38]. IL-10 is generally considered an anti-inflammatory cytokine that inhibits

inflammation and the synthesis of pro-inflammatory cytokines [39]. Considering these biological links, it is rational to surmise that the balance between inflammatory and anti-inflammatory factors such as MCP-1 and IL-10 is an important indicator of inflammation in the setting of nAMD [40,41]. Importantly, we found that these two cytokines were independently associated with BCVA besides CRT. It is generally believed that the increase in MCP-1 attracts ocular macrophages. leading to the digestion of the retinal pigment epithelium and Bruch's membrane [42]. Histologic examinations of surgical specimens showed macrophages located in regions with atrophy of the retinal pigment epithelium, Bruch's membrane breakdown, and CNV [43–47]. In contrast, IL-10 is believed to play a significant role in suppressing the formation of subretinal fibrosis [48]. IL-10 is involved in regulating the early and later movement of macrophages in and out of the injured nerve, lowers the expression of proinflammatory chemokines and cytokines, and plays a role in the inducing macrophages to shift from proinflammatory to anti-inflammatory via myelin-phagocytosis [49]. The results of the present study support the idea that MCP-1 and IL-10 control such different aspects of inflammation occurring in eyes with nAMD.

We also demonstrated that higher MCP-1 and lower CXCL13 levels at baseline were associated with a better change of BCVA at 12 months. Considering the aforementioned association showing that eyes with a higher MCP-1 concentration had a worse baseline BCVA, these eyes with higher MCP-1 concentrations at baseline had a better chance to improve BCVA, considering the ceiling effects of VA, which at least in part explains why higher MCP-1 concentrations at baseline were associated with a greater improvement of BCVA. CXCR5, the receptor for CXCL13, may be involved in protecting the retinal pigment epithelium and retinal cells during ageing, and its loss may lead to AMD-like pathological changes in aged mice [50]. However, the role of CXCL13 is less clear in the context of nAMD compared with other cytokines.

## Number of injections and cytokines

We assumed that the number of injections was an important parameter that reflects disease activity more than VA, because VA is also affected by numerous factors such as anatomical changes including the integrity of the external limiting membrane and the ellipsoid zone. Here, we showed that cytokine concentrations were strongly associated with the number of injections required for the first year, while clinical and demographic factors were not. Importantly, as shown in this study, other than MMP-9, aqueous humour proteins at baseline might not be associated with the number of injections; however, CXCL12 and IL-10 at 2 months may be useful to predict the number of ranibizumab injections after the initial loading dose. MMP-9 has a role in the control of the extracellular matrix. In the retinal pigment epithelium/choroid, MMP-9 is involved in the remodelling of Bruch's membrane and controls the permeability of the outer retinal barrier [51]. MMP-9 is considered to be involved in large pore formation in Bruch's membrane, which is critical for CNV to grow into the subretinal space [52]. Further studies including the balance of extracellular matrix remodelling factors such as TIMP/MMP [53] are required to understand how low concentrations of MMP-9 are associated with the number of injections. CXCL12 is a chemotactic factor for lymphocytes and promotes angiogenesis [54,55]. Interestingly, bone marrow-derived cells distributed in a mouse laser-induced CNV model are CXCL12 receptor-positive [56]. Therefore, the increased concentration of CXCL12 at 2 months may indicate the presence of residual subclinical inflammation/inflammatory cell accumulation after the loading dose. Notably, CXCL12 may be another important molecule involved in CNV activity, as suggested previously [57]. Interestingly, those patients with high IL-10 concentrations at 2 months required more frequent injections. This

may be in line with a recent hypothesis that a subpopulation of macrophages, referred to as alternatively activated macrophages, may be involved in nAMD [58]. Inflammation occurring in eyes with nAMD is mediated by several types of immune cell, among which myeloid cells (macrophages/monocytes) are considered to be of the utmost importance. Myeloid cells can be classified into at least two subpopulations, i.e. classically activated and alternatively activated monocytes/macrophages, which control two different aspects of inflammation; in general, the former exacerbates inflammation in the acute phase of inflammation, while the latter promotes angiogenesis in the chronic phase of inflammation, although this depends on the context in which inflammation occurs. IL-10 is produced by alternatively activated macrophages. Therefore, it is tempting to consider that the high levels of IL-10 are associated with an increased number of alternatively activated macrophages and aberrant angiogenesis, and because of such increased angiogenic drive, those patients with high levels of IL-10 at 2 months required an increased number of injections. In this aspect, MCP-1 is indispensable for classically activated macrophages, which accelerate inflammation, but not angiogenesis, and possibly leading to exudative changes. Although further studies are needed to validate our findings, our results underscore the importance of focusing on aqueous humour protein levels after loading injections for prognostication.

## Limitations

The current study has several limitations. The sample size was relatively small, only Japanese individuals were involved, and PCV and typical nAMD were mixed. The high heterogeneity of the studied samples as per AMD traits limits the study's conclusions. This study was also limited by a lack of a standardized and masked treatment protocol. The current results should be validated in other cohorts. However, we believe our results are still relevant considering that most of the patients adhered to the protocol presented and the *pro re nata* treatment regimen is one of the most commonly employed therapeutic approaches. Further studies are warranted to confirm our results.

## Conclusion

Higher MCP-1 and lower CXCL13 concentrations at first injection were predictor variables that were significantly associated better BCVA prognosis at 12 months. Higher concentrations of CXCL12 at the third injection were associated with a smaller number of ranibizumab injections. Additionally, IL-10 concentrations at 2 months was associated with the number of injections. Treatment outcomes can be predicted by investigating the concentrations of these aqueous humour proteins, which are potential biomarkers for precision medicine.

## Supporting information

**S1 Fig.**
(TIF)

**S1 Table. Factors associated with baseline characteristics.**
(DOCX)

**S2 Table. Factors associated with CRT and CCT changes at 2 months.**
(DOCX)

**S1 Text. Supplementary information of central retinal thickness and central choroidal thickness.**
(DOCX)

**S2 Text. Influence of fluorescein and indocyanine green to measurements using multiplex cytokine assay.**
(DOCX)

## Author Contributions

**Conceptualization:** Hidenori Takahashi, Yasuo Yanagi.

**Data curation:** Yusuke Arai, Hidenori Takahashi.

**Formal analysis:** Yusuke Arai, Hidenori Takahashi.

**Funding acquisition:** Hidenori Takahashi.

**Investigation:** Hidenori Takahashi.

**Methodology:** Hidenori Takahashi.

**Project administration:** Hidenori Takahashi.

**Supervision:** Hidetoshi Kawashima.

**Writing – original draft:** Yusuke Arai.

**Writing – review & editing:** Yusuke Arai, Hidenori Takahashi, Satoru Inoda, Xue Tan, Shinichi Sakamoto, Yuji Inoue, Yujiro Fujino, Hidetoshi Kawashima, Yasuo Yanagi.

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
