## [Decision Letter · Decision Letter 0]

17 Sep 2019

PONE-D-19-23157

Inflammatory cytokines and treatment outcomes of anti-VEGF therapy in neovascular age-related macular degeneration

PLOS ONE

Dear Dr. Takahashi,

Thank you for submitting your manuscript to PLOS ONE. After careful consideration, we feel that it has merit but does not fully meet PLOS ONE’s publication criteria as it currently stands. Therefore, we invite you to submit a revised version of the manuscript that addresses the points raised during the review process.

Both expert reviewers indicated that your submission was interesting but that the results do not fully support your conclusions.  Reviewer 1 indicated flaws in your methodology, particularly statistical analysis and reviewer 2 indicated that your present paper overlaps with your earlier paper, Sakamoto S, Takahashi H, Tan X, et al.

Br J Ophthalmol 2018;102:448–454.  I share this reviewers concerns that you may be presenting the same results in both papers, for example Table 2 in the earlier paper seems to present the same results as Table 1 in the current paper. In rewriting your paper for *PLoS One,  *please be clear what distinguishes the new paper from the older one.

We would appreciate receiving your revised manuscript by Nov 01 2019 11:59PM. To enhance the reproducibility of your results, we recommend that if applicable you deposit your laboratory protocols in protocols.io, where a protocol can be assigned its own identifier (DOI) such that it can be cited independently in the future. For instructions see: http://journals.plos.org/plosone/s/submission-guidelines#loc-laboratory-protocols

We look forward to receiving your revised manuscript.

Kind regards,

Alfred S Lewin, Ph.D.

Academic Editor

PLOS ONE

Journal Requirements:

1. We noticed you have some minor occurrence of overlapping text with the following previous publication(s), which needs to be addressed:

https://bjo.bmj.com/content/102/4/448

https://jamanetwork.com/journals/jamaophthalmology/fullarticle/426210

https://www.ncbi.nlm.nih.gov/pubmed/26674868

In your revision ensure you cite all your sources (including your own works), and quote or rephrase any duplicated text outside the methods section. Further consideration is dependent on these concerns being addressed.

2. Thank you for including your competing interests statement; "Dr Takahashi received lecturer’s fees from Kowa Pharmaceutical, Novartis Pharmaceuticals, Bayer Yakuhin, and Santen Pharmaceuticals, and grants from Bayer Yakuhin and Novartis Pharma, outside this work. Dr Inoue received lecturer’s fees from Kowa Pharmaceuticals, Novartis Pharmaceuticals, Bayer Yakuhin, and Santen Pharmaceuticals outside this work. Dr Kawashima received lecturer’s fees from Kowa Pharmaceutical, Novartis Pharmaceuticals, and Santen Pharmaceuticals outside this work. Dr Yanagi received lecturer’s fees and grants from Santen Pharmaceuticals outside this work. He is an advisory board member for Bayer Pharmaceuticals and a consultant for Santen Pharmaceuticals. Dr Arai, Dr Tan, Dr Inoda, Dr Sakamoto, and Dr Fujino declare no potential conflict of interest."

Reviewers' comments:

Reviewer's Responses to Questions

**Comments to the Author**

1. Is the manuscript technically sound, and do the data support the conclusions?

Reviewer #1: Partly

Reviewer #2: Partly

2. Has the statistical analysis been performed appropriately and rigorously? 

Reviewer #1: No

Reviewer #2: I Don't Know

3. Have the authors made all data underlying the findings in their manuscript fully available?

Reviewer #1: Yes

Reviewer #2: No

4. Is the manuscript presented in an intelligible fashion and written in standard English?

Reviewer #1: Yes

Reviewer #2: Yes

5. Review Comments to the Author

Reviewer #1: This a very interesting study on aqueous humour cytokines as biomarkers of nAMD treatment response as per VA and number of injections at 12 months. Nonetheless, shows a strong rationale and well-intended methodology. However, there are significant flaws in the Methods and Results section that, by instance, should not represent a major problem to solve. In addition, some information is lacking and some, by contrast, does not even need to be presented as it is later forgotten in the discussion. If most points are indeed faced and cleared it could then be ready to undergo a more detailed review.

Point by point considerations:

Abstract: consider a section-based writing instead such as Purpose-Methods-Results-Conclusion. Also, correlation coefficients mean association whereas beta coefficients mean prediction, which is not the same and therefore the conclusion should be rewritten.

Introduction: as a whole, we miss some more referenciated information on the local and systemic importance of inflammation for nAMD which, after all, is the rationale of the present study.

49: "AMD" use should be stated in the very first part of the text, not afterwards

54-55 and 57-58: consider softening such strong statements that don't even show a reference

60-64: explanation of the present study should not be present in the Introduction section

73-74: a reference is lacking in this statement

85: was this informed consent written or oral? Please specify

92: were non-retinal diseases such as glaucoma or anterior uveitis excluded?

95: how was this obtained? Through a clear corneal paracenthesis? On the slit lamp? Please specify

101-102: how where 0 values handled?

103-104: reference that statement

105: how were these vitreous samples obtained? on which basis and time? same subjects? on what inclusion criteria and randomisation?

106-107: need dispersion statistics such as standard deviation or IQR when a mean/median is showed

109: how were these 29 subjects chosen? on which basis?

114-115: does this mean ICG and FA was undertaken in all cases?

125: specify which ranibizumab concentration is used

126: "essentially as described in the CATT", please specify which deviation from CATT protocol this means

127: treatment decisions are therefore relied on the physician, with no protocol adjusted or blind assessment. Since this, conclusions as per visual acuity prognosis at 12 months are unfair as treatment is not defined nor controlled in the group.

129: specify which chart of vision is used

142: Pearson correlation demands a normality assumption, which is not clearly stated in the text, please update accordingly, also referring to used normality tests.

145: log transformation of which variables? cytokines? please specify

157: add dispersion statistics to AL mean

158-159: definition criteria for PCV and type of nAMD is not stated in the manuscript, please update

167-168: paired tests have not been claimed in the methods section, please update

170: are these means? medians? please state so; which statistical test is used?

182: where is this 2.7 coming from?

Table 2, 3, S1 and S2: it is not clear to me whether these are correlation coefficients, beta or standardized beta. Please update accordingly. Also a 95% CI should be informed for all data and a total R square in all tables.

201 and on: "better" is an unclear term, use "improvement" instead.

231: a Table 4 on this topic is also mandatory as previously done

252-255: even though S1/S2 tables are supplied, consider stating the main found coefficients in the manuscript in order to give a clear vision of their magnitudes

301-302: such bivariant statement is not further explained in the manuscript and could be forsaken.

303: a clear discussion on the reason why lower MCP-1 is associated to better BCVA at baseline and, at the same time, higher levels are linked to better BCVA at 12 months is mandatory. This is point is forgotten in the discussion.

314: baseline MMP-9 significant association is not discussed and even goes against this statement.

324: IL-10 results are not discussed even though is an important agent with also results regarding VA at 12 months.

325-327: these interesting results should be discussed. If the authors find CRT and CCT to fall beyond the manuscript agenda is should be better to ban this topic from the start of the presented investigation.

331: PCV/AMD categorization is a strong limitation of the study that tampers its conclusions; authors could consider, given such proportion, to also present all results with subgroup analysis between PCV+ and PCV- cases that could end up with interesting findings that could however still keep statistical significance.

332: authors should also refer to the high heterogeneity of the studied samples as per AMD traits as well as the lack of a standardized and blinded treatment protocol that limit the study conclusion, especially regarding VA and number of injections.

335: IL-10 and MMP-9 findings regarding number of injections are missing without reason.

338: "imaging parameters" have been lightly discussed in the manuscript, this could be skipped in the conclusion or otherwise enrich their previous discussion

Conclusion: I would recommend a complete rewriting focusing on how these results could influence present investigations and settle up future ones.

Reviewer #2: This study by Arai et al. reports intravitreal concentrations of CXCL-12, CXCL-13, IL-10, IP-10, IL-6, MCP-1, CCL-11, CXCL-1 and MMP-9 from 48 eyes of 48 patients. The study seems well conducted, although the paper could be more precise.

Overall comments:

- Some of the data is previously published—as I can tell, at least the entire table 1 has been previously published? I think the text should be very clear on what has already been published, preferably leave those results out of this study.

- The study describes inflammatory cytokines, although some are anti-inflammatory, and some are not cytokines at all?

- The list of measured markers are in different order in the abstract, in the text, and in the tables. It is a minor thing, but might help the reader to use the same order.

- Please state your reason for selecting these specific assays in the introduction.

Specific comments:

Page 3, line 62: Observations are made at baseline, after 2 months, and after 12 months. Later in the paper (e.g. page 10, line 204) you report results after 3 months? Later again (page 14, l. 254) reports measures at 2 months.

Page 3, line 66: define induction phase.

Page 4, line 77-80: Is already part of the method section.

Page 4, line 90: Patients received treatment – not eyes.

Page 4, line 91: refers to aforementioned centres, but these are not mentioned?

Regarding cytokine assays: Lacks information on how many measures were above/below detection range? And what where the CV values?

Page 5 line 103-106: Please rephrase/clarify sentence.

Page 5-6 line 116-118: “To identify PCV, A-Mode ultrasonography was used to measure axial length” This line is confusing, please rephrase.

Regarding clinical examinations: There is no clear definition of the diagnostic criterias.

Page 6 line 133-139: More suitable in the clinical examination-section.

Page 7, line 144-146: What models in what variable?

Page 7, line 158-160: refers to “typical AMD”, with no clear definition of this.

Table 2: Heading should probably read: Factors associated with baseline VA.

Table 2, and 3: It is unclear what selection of variables for the multivariate analysis is based on? In the subheading it states that P<0.05, but then why select sex (P=0.058) and CCT (P=0.42), but not IL-6 (P=0.018) in table 2?

Page 11, line 207-209: This line is confusing: BCVA change is associated with BCVA change?

Discussion: I am having much difficulties in distinguishing what results are from the previously published study, and what is new. Please only include the new findings to discussion and conclusion. A better use of space would be to discuss the potential impact of the relevant chemokines on CNV sustainability.

6. PLOS authors have the option to publish the peer review history of their article (what does this mean?). If published, this will include your full peer review and any attached files.

Reviewer #1: No

Reviewer #2: Yes: Marie Krogh Nielsen

---

## [Author Response · Author response to Decision Letter 0]

1 Dec 2019

Dear editor and reviewers,

Thank you for reviewing our manuscript for its submission to PLOS ONE. We hereby submit a revised version of the manuscript that addresses the point-by-point response raised during the review process.

Both expert reviewers indicated that your submission was interesting but that the results do not fully support your conclusions. Reviewer 1 indicated flaws in your methodology, particularly statistical analysis and reviewer 2 indicated that your present paper overlaps with your earlier paper, Sakamoto S, Takahashi H, Tan X, et al.Br J Ophthalmol 2018;102:448–454. I share these reviewers concerns that you may be presenting the same results in both papers, for example Table 2 in the earlier paper seems to present the same results as Table 1 in the current paper. In rewriting your paper for PLoS One, please be clear what distinguishes the new paper from the older one.

Thank you for pointing out the duplication. We have deleted all duplicated sections.

1. We noticed you have some minor occurrence of overlapping text with the following previous publication(s), which needs to be addressed:

https://bjo.bmj.com/content/102/4/448

https://jamanetwork.com/journals/jamaophthalmology/fullarticle/426210

https://www.ncbi.nlm.nih.gov/pubmed/26674868

In your revision ensure you cite all your sources (including your own works), and quote or rephrase any duplicated text outside the methods section. Further consideration is dependent on these concerns being addressed.

We have checked this and updated in this revision.

2. Thank you for including your competing interests statement; "Dr Takahashi received lecturer’s fees from Kowa Pharmaceutical, Novartis Pharmaceuticals, Bayer Yakuhin, and Santen Pharmaceuticals, and grants from Bayer Yakuhin and Novartis Pharma, outside this work. Dr Inoue received lecturer’s fees from Kowa Pharmaceuticals, Novartis Pharmaceuticals, Bayer Yakuhin, and Santen Pharmaceuticals outside this work. Dr Kawashima received lecturer’s fees from Kowa Pharmaceutical, Novartis Pharmaceuticals, and Santen Pharmaceuticals outside this work. Dr Yanagi received lecturer’s fees and grants from Santen Pharmaceuticals outside this work. He is an advisory board member for Bayer Pharmaceuticals and a consultant for Santen Pharmaceuticals. Dr Arai, Dr Tan, Dr Inoda, Dr Sakamoto, and Dr Fujino declare no potential conflict of interest."

We added the statement of sharing policies to the end of the COI statement and to the cover letter.

Reviewer #1: Partly

Reviewer #2: Partly

We have updated the Methods section in accordance with your comment.

3. Have the authors made all data underlying the findings in their manuscript fully available?

Reviewer #1: Yes

Reviewer #2: No

We have uploaded all data to the following site: https://doi.org/10.6084/m9.figshare.7126922.v1.

Reviewer #1: This a very interesting study on aqueous humour cytokines as biomarkers of nAMD treatment response as per VA and number of injections at 12 months. Nonetheless, shows a strong rationale and well-intended methodology. However, there are significant flaws in the Methods and Results section that, by instance, should not represent a major problem to solve. In addition, some information is lacking and some, by contrast, does not even need to be presented as it is later forgotten in the discussion. If most points are indeed faced and cleared it could then be ready to undergo a more detailed review.

Thank you for your comments. Overall, we toned down the strong statements in the Introduction. Additionally, the Methods and Results sections have been re-written to include additional details, as described below. We believe that we have carefully responded to all of your comments.

Abstract: consider a section-based writing instead such as Purpose-Methods-Results-Conclusion. 

We simply followed the authors’ guidelines of PLOS ONE; the sample Abstract in the authors’ guidelines does not include subheadings. However, we did structure the Abstract so that Purpose was first, followed by Methods and Results, and put the Conclusions last. However, we are not sure if we are allowed to write subheadings as you suggested and left it as it is. If EBM and AE allow, we are happy to accept your proposal to use subheadings in the Abstract.

Also, correlation coefficients mean association whereas beta coefficients mean prediction, which is not the same and therefore the conclusion should be rewritten.

We have corrected the points as you suggested. Overall, we used the term ‘predictive variables’ to describe what we have done correctly (page 18, lines 356–361)

Introduction: as a whole, we miss some more referenciated information on the local and systemic importance of inflammation for nAMD which, after all, is the rationale of the present study.

We have added 9 references related to the importance of local and systemic inflammation in nAMD to strengthen our hypothesis (refs 4 to 12). 

49: "AMD" use should be stated in the very first part of the text, not afterwards

We have carefully reviewed the manuscript and corrected all abbreviations as appropriate.

54-55 and 57-58: consider softening such strong statements that don't even show a reference

We have toned down these statements and added references as appropriate (page 3, lines 55–58).

60-64: explanation of the present study should not be present in the Introduction section

We have moved this explanation to the last part of the Introduction -(page 4, line 80-86).

73-74: a reference is lacking in this statement

We have added references (page 4, line 76). 

85: was this informed consent written or oral? Please specify

We have indicated that we received written informed consent from the patients (page 4, line 91).

92: were non-retinal diseases such as glaucoma or anterior uveitis excluded?

We excluded other ocular diseases such as glaucoma or anterior uveitis (page 5, lines 100).

95: how was this obtained? Through a clear corneal paracenthesis? On the slit lamp? Please specify

We obtained aqueous humour through a clear corneal paracentesis under an operating microscope. This has been clarified in the revised manuscript (page 5, lines 103–105).

101-102: how where 0 values handled?

We imputed half the lowest detection value for values lower than the detection limit (page 5, lines 110–111). 

103-104: reference that statement

We have added a reference. (page 5, line 113). 

105: how were these vitreous samples obtained? on which basis and time? same subjects? on what inclusion criteria and randomisation?

After the study protocol was approved by the Institutional Review Board, we initially collected vitreous humour samples from 5 cases. However, we decided not to proceed with further vitreous sampling thereafter because of safety concerns; we noted that the vitreous was stuck in the needle, and the procedure could potentially increase the risk of developing proliferation of vitreous. Moreover, the volume of vitreous humour that was collected by vitreous tap was very low. The 5 vitreous humour samples were all from nAMD eyes. 

106-107: need dispersion statistics such as standard deviation or IQR when a mean/median is showed

We have added standard deviation as you suggested. (page 5, lines 115–117).

109: how were these 29 subjects chosen? on which basis?

We were not able to determine VEGF concentrations in all patients because the sample volume was insufficient for measuring the concentrations of cytokines (multiplex analysis) and VEGF (enzyme-linked immunosorbent assay [ELISA]). We first subjected the samples to multiplex analysis and then the remaining samples were subjected to ELISA. Therefore, we were able to measure VEGF concentrations in only 29 (60%) cases prior to anti-VEGF therapy. Additionally, this was stated in our previous publication, and therefore, we cited our previous publication here.

114-115: does this mean ICG and FA was undertaken in all cases?

You are correct; all cases underwent fluorescein and indocyanine green angiography. This is our routine work-up for nAMD in our institution, because of the relatively high prevalence of PCV among nAMD cases in the region where the study was conducted (Japan) (page 5, lines 122–125).

125: specify which ranibizumab concentration is used

The ranibizumab concentration was 0.5 mg in 0.05 mL of solution (page 6, line 130). We believe this is the only available dosage outside the US.

126: "essentially as described in the CATT", please specify which deviation from CATT protocol this means

We have changed this line and detailed the PRN regimen we employed to describe the protocol clearly.

127: treatment decisions are therefore relied on the physician, with no protocol adjusted or blind assessment. Since this, conclusions as per visual acuity prognosis at 12 months are unfair as treatment is not defined nor controlled in the group.

This is a limitation of the present study. 

129: specify which chart of vision is used

We measured best corrected visual acuity (BCVA) using a 5 M Landolt C VA chart.

142: Pearson correlation demands a normality assumption, which is not clearly stated in the text, please update accordingly, also referring to used normality tests.

A normal distribution was confirmed with Shapiro-Wilk’s W-test (page 7, lines 151–153). 

145: log transformation of which variables? cytokines? please specify

We added “of aqueous humour proteins” (page 7, line 159).

157: add dispersion statistics to AL mean

We have rewritten and updated the Results accordingly.

158-159: definition criteria for PCV and type of nAMD is not stated in the manuscript, please update

We have rewritten and updated the Results accordingly.

167-168: paired tests have not been claimed in the methods section, please update

We have rewritten and updated the Methods for statistical analysis and have described all tests performed in this paper. 

170: are these means? medians? please state so; which statistical test is used?

Since the concentrations had a lognormal distribution, they are geometric means. We originally included Table 1 to clarify the demographics of the patients, but this was the same as in Table 2 of our previous study (Sakamoto et al., BJO 2018) and we have deleted the entire table and put the necessary information in the Methods.

182: where is this 2.7 coming from?

By 2.7 (=e), we meant Napier’s constant, because we used natural logarithms. However, we have changed this to “per logMCP-1” to avoid any confusion.

Table 2, 3, S1 and S2: it is not clear to me whether these are correlation coefficients, beta or standardized beta. Please update accordingly. Also a 95% CI should be informed for all data and a total R square in all tables.

We have updated the tables according to your comments. However, adding 95% confidence intervals (CIs) made the tables too complex. Most guidelines recommend including at least a P value, and not a 95% CI, because it provides sufficient information to assess the significance of the value. R2 values were stated for all multivariate analysis.

201 and on: "better" is an unclear term, use "improvement" instead.

We have corrected this.

231: a Table 4 on this topic is also mandatory as previously done

Sorry, but we do not have a Table 4; we carefully reviewed and corrected all tables.

252-255: even though S1/S2 tables are supplied, consider stating the main found coefficients in the manuscript in order to give a clear vision of their magnitudes

We chose not to include this topic in the revised manuscript and all supplementary tables have been deleted.

301-302: such bivariant statement is not further explained in the manuscript and could be forsaken.

We have deleted this sentence.

303: a clear discussion on the reason why lower MCP-1 is associated to better BCVA at baseline and, at the same time, higher levels are linked to better BCVA at 12 months is mandatory. This is point is forgotten in the discussion.

Lower MCP-1 concentrations were associated with better BCVA at baseline and, at the same time, higher levels were linked to a better BCVA ‘gain’ at 12 months. We have added ‘The current results demonstrated that eyes with higher MCP-1 concentrations had worse baseline BCVA compared to those with lower MCP-1 concentrations. Those with higher MCP-1 concentrations at baseline had a better chance of BCVA gain compared to those with lower MCP-1 concentrations, which is probably because of the low baseline BCVA of this group, at least in part’.

314: baseline MMP-9 significant association is not discussed and even goes against this statement.

We have added ‘Moreover, the loss of structural recuperative power will necessitate repeated injections’.

324: IL-10 results are not discussed even though is an important agent with also results regarding VA at 12 months.

The Discussion section has been re-written and we have added a discussion about the results of IL-10. Inflammation occurring in eyes with nAMD is mediated by several types of immune cell, among which myeloid cells (macrophages/monocytes) are considered to be of the utmost important. Laboratory studies suggest that myeloid cells can be classified into at least two subpopulations, namely, classically activated and alternatively activated monocytes/macrophages, which control two different aspects of inflammation; in general, the former exacerbates inflammation in the acute phase of inflammation, while the latter exacerbates angiogenesis in the chronic phase of inflammation, although there is some debate on this issue. IL-10 is produced by alternatively activated macrophages, which contribute to angiogenesis. It is tempting to consider that in addition to exudative changes, low levels of IL-10 are associated with the aberrant control of angiogenesis, which may affect the number of injections. 

325-327: these interesting results should be discussed. If the authors find CRT and CCT to fall beyond the manuscript agenda is should be better to ban this topic from the start of the presented investigation.

We have deleted these results and focused instead on the factors associated with treatment outcome in the current manuscript. We decided that CRT and CCT are two important topics that will be analysed and discussed in a separate paper in the future.

331: PCV/AMD categorization is a strong limitation of the study that tampers its conclusions; authors could consider, given such proportion, to also present all results with subgroup analysis between PCV+ and PCV- cases that could end up with interesting findings that could however still keep statistical significance.

We agree with your opinion; therefore, in our current analysis, we tried to elucidate the effects of disease type (CNV with or without PCV), and included this as a covariant in stepwise variant selection. However, disease type was selected only as a predictor variable of BCVA change and in no other situations.

332: authors should also refer to the high heterogeneity of the studied samples as per AMD traits as well as the lack of a standardized and blinded treatment protocol that limit the study conclusion, especially regarding VA and number of injections.

We have revised the Limitations section accordingly. However, we believe our results are still relevant considering that most of the patients adhered to the protocol presented and the treatment regimen was based on one of the most commonly employed therapeutic approaches.

335: IL-10 and MMP-9 findings regarding number of injections are missing without reason.

We have now discussed this in the revised manuscript.

338: "imaging parameters" have been lightly discussed in the manuscript, this could be skipped in the conclusion or otherwise enrich their previous discussion

We have deleted this from the manuscript.

Conclusion: I would recommend a complete rewriting focusing on how these results could influence present investigations and settle up future ones.

Thank you for your constructive suggestions. We have added a section summarizing how the current investigation builds on previous studies in this field and how it impacts on future directions. ‘Our results clearly demonstrated that different aqueous humour proteins, including cytokines associated with classically and alternatively activated macrophages, may be involved in AMD in a different way. Part of this could be ascribed to the fact that each factor is involved in different aspects of inflammation, i.e. exudation and angiogenesis. Further studies focusing on different subclasses of immune cells and their markers are warranted for better clinical care. 

Reviewer #2: This study by Arai et al. reports intravitreal concentrations of CXCL-12, CXCL-13, IL-10, IP-10, IL-6, MCP-1, CCL-11, CXCL-1 and MMP-9 from 48 eyes of 48 patients. The study seems well conducted, although the paper could be more precise.

Overall comments:

- Some of the data is previously published—as I can tell, at least the entire table 1 has been previously published? I think the text should be very clear on what has already been published, preferably leave those results out of this study.

We have deleted the replicated data from the revised manuscript.

- The study describes inflammatory cytokines, although some are anti-inflammatory, and some are not cytokines at all?

You are correct. We measured the aqueous humour concentrations of several proteins, not only cytokines; therefore, we changed the terminology to more accurately describe what was measured.

- The list of measured markers are in different order in the abstract, in the text, and in the tables. It is a minor thing, but might help the reader to use the same order.

We have re-ordered this and rewritten the Abstract, text, and tables.

- Please state your reason for selecting these specific assays in the introduction.

We have added the reason underlying the selection of these assays.

Specific comments:

Page 3, line 62: Observations are made at baseline, after 2 months, and after 12 months. Later in the paper (e.g. page 10, line 204) you report results after 3 months? Later again (page 14, l. 254) reports measures at 2 months.

We measured at 2 months (at the 3rd injection). We apologise for the confusion and have corrected this mistake.

Page 3, line 66: define induction phase.

We have added ‘initial 3 monthly consecutive injection phase’. (Page 3, lines 69–70).

Page 4, line 77-80: Is already part of the method section.

We have deleted the duplicated section.

Page 4, line 90: Patients received treatment – not eyes.

We have corrected this.

Page 4, line 91: refers to aforementioned centres, but these are not mentioned?

We recruited patients from two institutions. This sentence did not make any sense at all and we have corrected the mistake. 

Regarding cytokine assays: Lacks information on how many measures were above/below detection range? And what where the CV values?

No proteins were above the detection range. The number of samples below the detection range, number of samples below the detection limit, and coefficient of variation (CV) of CXCL1, IP-10, CXCL12, CXCL13, MCP-1, CCL11, IL-6, IL-10, and MMP-9 were 1, 0, 3.6%, 0, 0, 3.8%, 0, 0, 6.7%, 0, 0, 4.6%, 0, 0, 2.1%, 0, 0, 4.7%, 0, 0, 2.0%, 68, 37, 4.5%, and 17, 1, 3.4%, respectively. Most of the samples were within the detection range, except for MMP-9.

Page 5 line 103-106: Please rephrase/clarify sentence.

We rephrased and clarified the sentence on page 5, lines 111–113.

Page 5-6 line 116-118: “To identify PCV, A-Mode ultrasonography was used to measure axial length” This line is confusing, please rephrase.

We apologise for our mistake. We have corrected the sentences. ‘To identify PCV…’ was an infinitive clause for the previous sentence.

Regarding clinical examinations: There is no clear definition of the diagnostic criterias.

We apologise for the lack of diagnostic criteria. We added references for the diagnosis of PCV and RAP to page 6, line 122.

Page 6 line 133-139: More suitable in the clinical examination-section.

We moved this paragraph to the clinical examination section on page 6, lines 139–145.

Page 7, line 144-146: What models in what variable?

We have revised the statistical analysis section.

Page 7, line 158-160: refers to “typical AMD”, with no clear definition of this.

We have revised the Results section.

Table 2: Heading should probably read: Factors associated with baseline VA.

We have changed the heading of Table 1 as you suggested (page 8 line 189).

Table 2, and 3: It is unclear what selection of variables for the multivariate analysis is based on? In the subheading it states that P<0.05, but then why select sex (P=0.058) and CCT (P=0.42), but not IL-6 (P=0.018) in table 2?

We apologise for the confusion. To clarify this, we have rewritten the Abstract and Methods to describe the methods used. We used stepwise variable selection, described on page 8, lines 158–159, to build the best prediction model that minimizes the Bayesian Information Criterion. This is an objective method that automatically determines which variables should be included in the model. We used the forward (increasing) method. 

Page 11, line 207-209: This line is confusing: BCVA change is associated with BCVA change?

We apologize for the confusion. The larger change in BCVA at 2 months was associated with an increased change of BCVA of 12 months. We deleted the sentence previously on page 12, lines 222–224.

Discussion: I am having much difficulties in distinguishing what results are from the previously published study, and what is new. Please only include the new findings to discussion and conclusion. A better use of space would be to discuss the potential impact of the relevant chemokines on CNV sustainability.

Although we initially thought that including such information was useful for the readers of the current manuscript, we have revised the text to delete the duplicated results (Table 1) and other previous findings to avoid concerns of possible duplicate publication. The Discussion was also re-written to focus more on the new findings of the current study (deleted lines from the Discussion, previously on page 17, lines 286–290).

---

## [Decision Letter · Decision Letter 1]

19 Dec 2019

PONE-D-19-23157R1

Aqueous humour proteins and treatment outcomes of anti-VEGF therapy in neovascular age-related macular degeneration

PLOS ONE

Dear Dr. Takahashi,

Thank you for submitting your manuscript to PLOS ONE. After careful consideration, we feel that it has merit but does not fully meet PLOS ONE’s publication criteria as it currently stands. Therefore, we invite you to submit a revised version of the manuscript that addresses the points raised during the review process.

Two reviewers evaluated this revised manuscript and both agreed that you have addressed the issues raised previously. Thank you for your efforts.  Both have made some suggestions for additional clarification of your findings. Please address these additional comments.

We would appreciate receiving your revised manuscript by Feb 02 2020 11:59PM. To enhance the reproducibility of your results, we recommend that if applicable you deposit your laboratory protocols in protocols.io, where a protocol can be assigned its own identifier (DOI) such that it can be cited independently in the future. For instructions see: http://journals.plos.org/plosone/s/submission-guidelines#loc-laboratory-protocols

We look forward to receiving your revised manuscript.

Kind regards,

Alfred S Lewin, Ph.D.

Academic Editor

PLOS ONE

Reviewers' comments:

Reviewer's Responses to Questions

**Comments to the Author**

1. If the authors have adequately addressed your comments raised in a previous round of review and you feel that this manuscript is now acceptable for publication, you may indicate that here to bypass the “Comments to the Author” section, enter your conflict of interest statement in the “Confidential to Editor” section, and submit your "Accept" recommendation.

Reviewer #1: All comments have been addressed

Reviewer #3: All comments have been addressed

2. Is the manuscript technically sound, and do the data support the conclusions?

Reviewer #1: Yes

Reviewer #3: Yes

3. Has the statistical analysis been performed appropriately and rigorously? 

Reviewer #1: Yes

Reviewer #3: Yes

4. Have the authors made all data underlying the findings in their manuscript fully available?

Reviewer #1: Yes

Reviewer #3: Yes

5. Is the manuscript presented in an intelligible fashion and written in standard English?

Reviewer #1: Yes

Reviewer #3: Yes

6. Review Comments to the Author

Reviewer #1: I do congratulate the authors for their job improving the manuscript, which is now technically sound and expressively strong. I can only point out two minor questions to undertake.

Lines 118-120: clarify whether VEGF in these 29 samples was vitreous or aqueous humour. Moreover, since not all samples were analyzed for VEGF search (as stated in your response), it ends up behaving as a huge bias for the study. Some could even think that the authors choose which samples they finally analyzed. I feel it is not enough to state a reference to back up such reasoning and I would rather include a detailed explanation, like the one that has been provided in the response, in the manuscript.

Conclusion: lacks a statement on IL-10 at 2 months associated with the number of injections, which has been extensively tackled in the Discussion, and should be at least briefly commented here.

Reviewer #3: In this paper Arai et al. investigated the relationship between Aqueous humour proteins and anti-VEGF therapy outcomes in nAMD. The authors have addressed previous comments. I have two additional minor comments:

1. Abstract: The authors state in the final sentence that "In conclusion, aqueous humour protein concentrations may be good predictors of BCVA change over 12 months and the number of injections in pro re nata treatment of exudative nAMD.". This is not my impression from this study - this study can only conclude that aqueous humor proteins may have predictive abilities - to determine whether they are good predictors, further biomarker analyses need to be performed.

2. Regarding the limitations of this study. This is perhaps in the lines of my comment #1. A limitation of this study is the lack of another independent sample, wherein the potentially predictive abilities of aqueous humor proteins can be tested. This approach allows concluding whether these proteins truly possess predictive abilities and how good they are in doing so.

3. Regarding methods: To my knowledge, the Procarta® Cytokine Assay Kit is a multiplex cytokine assay that is read using a fluorescence-based method. Do you have any insight from a pilot study or any investigation into whether fluorescein or indocyanine green (which is used in the retinal angiography in relation to diagnosis) influences measurements obtained from the Procarta® Cytokine Assay Kit? If not, then I think it would be worth mentioning that this is a potential source of bias and thus a limitation that must be kept in mind when interpreting the results of this study. Also, it is unclear in the manuscript when you did the sampling. Immediately after the diagnosis? Immediately prior to the injection? Immediately after the injection? Was it prior to the injection in baseline, but after at the first follow-up? These factors are important details with potential influence to the results. These factors should be outlined in a clear fashion.

7. PLOS authors have the option to publish the peer review history of their article (what does this mean?). If published, this will include your full peer review and any attached files.

Reviewer #1: No

Reviewer #3: No

---

## [Author Response · Author response to Decision Letter 1]

2 Feb 2020

Reviewer #1: I do congratulate the authors for their job improving the manuscript, which is now technically sound and expressively strong. I can only point out two minor questions to undertake.

Lines 118-120: clarify whether VEGF in these 29 samples was vitreous or aqueous humour. Moreover, since not all samples were analyzed for VEGF search (as stated in your response), it ends up behaving as a huge bias for the study. Some could even think that the authors choose which samples they finally analyzed. I feel it is not enough to state a reference to back up such reasoning and I would rather include a detailed explanation, like the one that has been provided in the response, in the manuscript.

First, these 29 samples were aqueous humour. Second, we have added explanations as you suggested. (lines 127-129)

Conclusion: lacks a statement on IL-10 at 2 months associated with the number of injections, which has been extensively tackled in the Discussion, and should be at least briefly commented here.

We have updated the conclusion section in accordance with your comment. (lines 380-381)

Reviewer #3: In this paper Arai et al. investigated the relationship between Aqueous humour proteins and anti-VEGF therapy outcomes in nAMD. The authors have addressed previous comments. I have two additional minor comments:

1. Abstract: The authors state in the final sentence that "In conclusion, aqueous humour protein concentrations may be good predictors of BCVA change over 12 months and the number of injections in pro re nata treatment of exudative nAMD.". This is not my impression from this study - this study can only conclude that aqueous humor proteins may have predictive abilities - to determine whether they are good predictors, further biomarker analyses need to be performed.

We have updated the abstract section in accordance with your comment. (line 45)

2. Regarding the limitations of this study. This is perhaps in the lines of my comment #1. A limitation of this study is the lack of another independent sample, wherein the potentially predictive abilities of aqueous humor proteins can be tested. This approach allows concluding whether these proteins truly possess predictive abilities and how good they are in doing so.

We agree with this. We have revised the limitation section. (lines 369-370)

3. Regarding methods: To my knowledge, the Procarta® Cytokine Assay Kit is a multiplex cytokine assay that is read using a fluorescence-based method. Do you have any insight from a pilot study or any investigation into whether fluorescein or indocyanine green (which is used in the retinal angiography in relation to diagnosis) influences measurements obtained from the Procarta® Cytokine Assay Kit? If not, then I think it would be worth mentioning that this is a potential source of bias and thus a limitation that must be kept in mind when interpreting the results of this study. Also, it is unclear in the manuscript when you did the sampling. Immediately after the diagnosis? Immediately prior to the injection? Immediately after the injection? Was it prior to the injection in baseline, but after at the first follow-up? These factors are important details with potential influence to the results. These factors should be outlined in a clear fashion.

We performed a preliminary test, and confirmed that the fluorescein and indocyanine green did not affect the measurement of cytokine levels in this Cytokine Assay Kit. We have added the results as S2-text for your review. In the text we mentioned “dye angiography did not affect the measurements (data not shown).” (lines 123 to 124)

---

## [Editor Report · Decision Letter 2]

5 Feb 2020

Aqueous humour proteins and treatment outcomes of anti-VEGF therapy in neovascular age-related macular degeneration

PONE-D-19-23157R2

Dear Dr. Takahashi,

We are pleased to inform you that your manuscript has been judged scientifically suitable for publication and will be formally accepted for publication once it complies with all outstanding technical requirements.

With kind regards,

Alfred S Lewin, Ph.D.

Section Editor

PLOS ONE
---

## [Editor Report · Acceptance letter]

14 Feb 2020

PONE-D-19-23157R2 

Aqueous humour proteins and treatment outcomes of anti-VEGF therapy in neovascular age-related macular degeneration 

Dear Dr. Takahashi:

I am pleased to inform you that your manuscript has been deemed suitable for publication in PLOS ONE. Congratulations! Your manuscript is now with our production department. 

With kind regards,

on behalf of

Dr. Alfred S Lewin 

Section Editor

PLOS ONE